# Combining Auxiliary Losses for Safer and More Robust Trajectory Prediction

## Abstract

Accurate trajectory prediction is essential for the safety and reliability of autonomous systems. Despite recent progress, models still struggle with scene compliance, often producing off-road or traffic-violating forecasts. We revisit and enhance three intuitive auxiliary objectives—Offroad Loss, Direction Consistency Loss, and Diversity Loss—that enhance map adherence, traffic rule compliance, and trajectory coverage. While each improves a specific aspect, our key finding is that only their combination delivers robust road-compliant predictions. To make this practical, we propose a lightweight adaptive weighting scheme that balances auxiliary losses automatically, succeeding where existing multi-task training strategies fail. Extensive experiments on nuScenes and Argoverse 2 show consistent gains in safety and robustness without sacrificing accuracy, with 43% decrease in off-road errors on average. Notably, under the SceneAttack benchmark, which perturbs road geometry to create out-of-distribution driving scenarios, our method reduces off-road errors by 25%, demonstrating that learned road compliance transfers to unseen environments. Our plug-and-play package can be integrated into any trajectory predictor, and code will be released.

## 1 Introduction

Trajectory prediction plays a central role in autonomous systems, directly affecting the safety and reliability of self-driving vehicles. Recent advances have enabled multimodal prediction models that account for driver intent and uncertainty. Yet, models still struggle with scene compliance—they may predict trajectories that leave the road, violate traffic flow, or miss plausible maneuvers in complex settings (Figure 1). Prior work has shown that such models often lack sufficient map understanding, failing to adapt even to naturalistic road perturbations Bahari et al. (2022). Improving scene awareness therefore remains a key challenge.

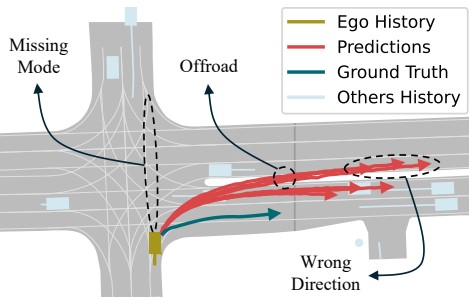

Figure 1: Trajectory predictions by Wayformer Nayakanti et al. (2022), a well-established model, highlighting errors such as off-road movements, moving against traffic, and missed predictions for other plausible maneuvers at the intersection. Our training package corrects such errors.

A major reason for these failures is that standard training losses are accuracy-focused: they optimize only the trajectory closest to ground truth and provide little incentive to avoid traffic violations or cover diverse behaviors. To address this, we design three refined auxiliary objectives—Offroad Loss, Direction Consistency Loss, and Diversity Loss—that guide predictions toward drivable areas,

respect road directions, and capture multiple feasible maneuvers. While each objective addresses one weakness, their combination is essential: only together do they consistently improve scene awareness, diversity, and compliance across all prediction modes.

Balancing such objectives is challenging, as fixed weights require costly hyperparameter tuning and may destabilize training. We therefore introduce a simple adaptive weighting scheme that adjusts each auxiliary loss on the fly based on its interaction with the main objective. This makes the package easy to integrate and ensures that auxiliary terms contribute only when beneficial.

Our contributions are threefold:

1. We propose a unified auxiliary loss package that strengthens scene compliance by combining off-road, direction, and diversity objectives.

2. We develop a lightweight adaptive weighting strategy that succeeds where established multi-task training methods fail.

3. We demonstrate that improved road compliance transfers to out-of-distribution scenarios, reducing off-road errors by 25% on the SceneAttack benchmark Bahari et al. (2022), in addition to a 43% reduction on standard nuScenes and Argoverse 2 evaluations.

These results highlight that the key lies not in designing entirely new losses, but in effectively combining and weighting existing ones. Our approach improves robustness and generalization in trajectory prediction without sacrificing accuracy, providing a plug-and-play solution for modern forecasting models.

## 2 RELATED WORK

**Vehicle Trajectory Prediction** Early approaches rasterized bird's-eye-view inputs and used CNN backbones to forecast motion Ren et al. (2021); Biktairov et al. (2020); Chai et al. (2019); Hong et al. (2019); Casas et al. (2018). While effective, rasterization adds memory/computation overhead and burdens the network with extracting 2D geometry already available in maps. Vectorized map encodings alleviated these issues via GNNs and sequence models Liang et al. (2020); Gao et al. (2020); Ha & Jeong (2023); Li et al. (2022); Park et al. (2020); Chiara et al. (2022); Lin et al. (2022); Ip et al. (2021). Transformers further improved long-range interactions and multi-agent reasoning Nayakanti et al. (2022); Girgis et al. (2022); Liu et al. (2024); Shi et al. (2022); Huang et al. (2023). Despite architectural progress, scene compliance remains challenging: models still produce off-road or wrong-way forecasts and may miss valid modes at intersections, as also highlighted by SceneAttack's naturalistic road perturbations Bahari et al. (2022).

**Prediction quality improvement** Prior works hard-wire scene structure (e.g., lane graphs or frenet wrappers) Deo et al. (2022); Gilles et al. (2021); Gu et al. (2021); Hallgarten et al. (2023) and promote higher multimodal diversity Wang et al. (2022); Kim et al. (2023); Park et al. (2020). Complementary to architecture changes, several papers define off-road compliance metrics and sometimes use them as auxiliary losses Ridel et al. (2020); Niedoba et al. (2019); Boulton et al. (2020); Messaoud et al. (2020); Chang et al. (2019); Cui et al. (2021). These typically rely on raster masks; we instead use a vectorized signed-distance formulation that is more precise for modern transformer predictors. Heading-alignment ("yaw") regularizers have also been explored Greer et al. (2021), but often exclude intersections or rely on nearest-centerline heuristics; our direction loss considers all candidate centerlines and balances position/heading consistency, avoiding special-case exclusions. Reinforcement-learning formulations exist Casas et al. (2020), but suffer from training instability; our objectives are fully differentiable and low-overhead.

**Adaptive loss weighting** Tuning weights for multiple objectives is challenging and often unstable. Uncertainty-based weighting Kendall et al. (2018) is effective for multi-task heads, but less suited when auxiliary terms should not harm main-task accuracy - which is our case. Gradient-based methods scale or reshape contributions using gradient magnitudes and conflicts: GradNorm balances by norm Chen et al. (2018); PCGrad projects away conflicting components Yu et al. (2020); CAGrad trades off average improvement with worst-task guarantees Liu et al. (2021); and Nash-MTL views combination as a bargaining game Navon et al. (2022). More recent task-aware schedules (e.g.,

AdaTask Yang et al. (2023)) adjust per-task learning dynamics rather than static weights. We adopt a simple scheme that jointly accounts for (i) gradient-norm balance and (ii) cosine-similarity alignment with the main loss, and we show that established alternatives (e.g., pure cosine gating Du et al. (2018) and pure norm scaling Chen et al. (2018)) underperform on trajectory prediction, where auxiliary terms must help without degrading accuracy.

# 3 METHOD

This section formally defines the vehicle trajectory prediction problem to establish a mathematical framework. It then reviews the commonly used accuracy-based training approaches in recent models and discusses their limitations. Next, we introduce three auxiliary loss functions designed to enrich models with greater scene understanding and promote diversity in predictions to overcome previous limitations. Finally, we present our adaptive loss weighting strategy, which dynamically adjusts the contribution of these losses to ensure they improve robustness and safety without compromising model accuracy.

## 3.1 FORMULATION

Consider the trajectory prediction problem involving an ego agent surrounded by $N$ neighbouring agents within a scene. Let $s_t^i = (x_t^i, y_t^i)$ define the state of agent $i$ at timestep $t$. We have access to the observed trajectories $\boldsymbol{x}^i = [s_{-t}^i, \cdots, s_{-1}^i, s_0^i]$ for all agents over the past $t$ timesteps, aggregated as $\boldsymbol{x} = [\boldsymbol{x}^0, \boldsymbol{x}^1, \cdots, \boldsymbol{x}^N] \in \mathbb{R}^{N \times t \times 2}$. Additionally, the input includes the drivable area $\Omega$, represented as a union of several closed polygons, and the centerlines $\boldsymbol{c} \in \mathbb{R}^{S \times K \times 3}$, with each of $S$ centerline segments containing $K$ points characterized by their 2D position and yaw $(x, y, \theta)$. The prediction model $f$ aims to predict the ego agent's ground truth trajectory $\hat{\boldsymbol{y}} = [s_1^0, \cdots, s_T^0]$ over the next $T$ timesteps. Given the multimodal nature of the prediction task, stemming from uncertainties in drivers' intentions, the model $f$ generates $M$ distinct possible trajectories, thus $\boldsymbol{y} = f(\boldsymbol{x}, \Omega, \boldsymbol{c}) \in \mathbb{R}^{M \times T \times 2}$.

## 3.2 PREVIOUS OBJECTIVE FUNCTIONS

Multimodal predictions are a fundamental aspect of current trajectory prediction models, with each model generating $M$ distinct trajectories. Traditionally, these models are trained using objective functions like minADE, defined as:

$$\text{minADE} = \min_{1 \leq m \leq M} \frac{1}{T} \sum_{i=1}^{T} \|\boldsymbol{y}_t^m - \hat{\boldsymbol{y}}_t^m\|_2. \tag{1}$$

This function focuses on minimizing the error for the trajectory closest to the ground truth, effectively pushing it closer to the current outcome while allowing other predictions to represent alternative possible paths. While these accuracy-based objective functions are effective in guiding models towards higher precision predictions, they do not necessitate scene-awareness and safety. Moreover, in such loss functions, only the closest trajectory to the ground-truth receives gradient updates during training, leaving the quality of other predicted trajectories largely unrefined.

## 3.3 PROPOSED OBJECTIVE FUNCTIONS

To address the limitations of the common accuracy-based training objectives, we propose three new loss functions. Unlike traditional methods, these functions supervise all prediction modes, enhancing different aspects of the model's performance. Each function is designed to infuse the model with deeper knowledge, targeting specific shortcomings in existing approaches. Figure 2 illustrates these loss functions.

### 3.3.1 OFFROAD LOSS

The first proposed loss function directs predictions toward the drivable area and penalizes off-road deviations by employing a signed distance function between the predicted trajectory $\boldsymbol{y}$ and the drivable area $\Omega$. This function is continuous and differentiable, with its gradient directed toward the nearest point within $\Omega$. Figure 2a visually demonstrates the offroad function, where the penalty

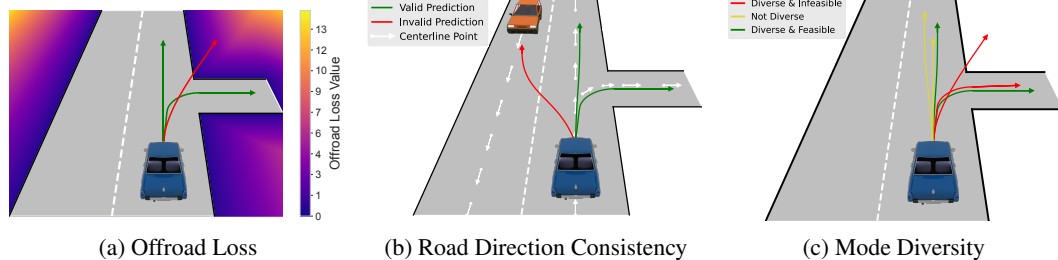

(a) Offroad Loss       (b) Road Direction Consistency       (c) Mode Diversity

Figure 2: An illustration for our proposed loss functions. The colors in (a) show the Offroad Loss values for areas around the blue vehicle, with the red offroad trajectory having a high penalty. Panel (b) illustrates Road Direction Consistency, showing centerline points and directions; the incorrect red trajectory fails to align with the proper road direction. In (c), we compare three prediction sets: red trajectories demonstrate diversity but are infeasible due to straying off the drivable area; yellow trajectories are feasible but lack diversity, missing a potential right turn; green trajectories successfully combine diversity with feasibility, accurately reflecting viable path options.

values are depicted across different areas around the vehicle. Mathematically, the signed distance function $\phi$ for $\Omega$ is defined such that it returns the shortest distance from any point $x$ to the boundary $\partial\Omega$ of $\Omega$, being negative if $x$ is inside $\Omega$ and positive otherwise. We define our Offroad loss function as:

$$\text{Offroad Loss}(\boldsymbol{y}, \Omega) = \frac{1}{M} \sum_{i=1}^{M} \sum_{t=1}^{T} \max(\phi(\boldsymbol{y}_t^i, \Omega) + m, 0), \tag{2}$$

where $\boldsymbol{y}_t^i$ represents the position at timestep $t$ of the $i$th prediction, $\phi$ is the signed distance function, and $m$ is a margin that maintains a buffer from $\Omega$'s boundary. This loss sums across all prediction points, with zero indicating presence within the drivable area (including a margin of $m$ meters) and increasing as predictions move further from $\Omega$.

To compute the signed distance function $\phi(\boldsymbol{p}, \Omega)$, we calculate the distance from point $\boldsymbol{p}$ to all polygon edges in $\Omega$ and select the minimum distance to determine the closest boundary. For determining whether $\boldsymbol{p}$ is inside $\Omega$, a ray casting algorithm is utilized, where a ray extending from $\boldsymbol{p}$ along the x-axis is used to count the intersections with the polygon's edges. Then, point $\boldsymbol{p}$ is in or out of $\Omega$ if the number of crossings is odd or even, respectively. This method, commonly used for point-in-polygon tests, is adapted from computational geometry principles outlined by O'Rourke (1998). All computations are efficiently implemented on the GPU to minimize the performance overhead and accelerate the loss evaluation process. A pseudo algorithm could be found in §D.

### 3.3.2 DIRECTION CONSISTENCY ERROR

Given that road centerlines are part of the map context for prediction models, maintaining the correct directional alignment is crucial. To address instances where models may align with these centerlines in the incorrect direction, we propose the Road Direction Consistency loss function. As Figure 2b illustrates, this function imposes a high penalty on trajectories deviating from corresponding centerline's direction, ensuring alignment not just in position but also in orientation.

To implement it, we calculate the heading direction $\gamma_t^i$ of each predicted trajectory $\boldsymbol{y}^i$ at timestep $t$. The difference $\delta(c, y_t^i)$ between a centerline point $c = (x, y, \theta)$ and a trajectory point $y_t^i = (x', y', \gamma_t^i)$ is defined as:

$$\delta(c, y_t^i) = \max\left(\|(x, y) - (x', y')\|_2 - m_d, 0\right)$$
$$+ \max\left(|\theta - \gamma_t^i| - m_\theta, 0\right),$$

where $m_d$ and $m_\theta$ represent the allowable distance and angle margins, respectively.

The Road Direction Consistency loss then aggregates the minimum $\delta$ value between each trajectory point and all centerline points, summed across all trajectory points:

$$\text{Direction}(\boldsymbol{y}, c) = \frac{1}{M} \sum_{i=1}^{M} \sum_{t=1}^{T} \min_{\substack{1 \leq s \leq S \\ 1 \leq k \leq K}} \delta(c_k^s, y_t^i). \tag{3}$$

The flexibility of this loss function is particularly important in complex environments like intersections, where many centerlines are close together but may have very different directions. Matching the trajectory strictly to the nearest centerline, as Greer et al. (2021) does, can sometimes lead to incorrect alignments. Our approach allows for matching with any centerline while introducing a penalty for the distance, ensuring a more accurate match. This means the model might align a trajectory with a centerline that is not the closest one but has a more suitable heading. This method ensures each trajectory point is evaluated against the most appropriate centerline, effectively improving accuracy in both position and direction, especially in challenging scenarios.

### 3.3.3 MODE DIVERSITY

Ensuring a diverse set of predictions is critical for robust and safe route planning, particularly to capture all highly probable and plausible trajectories. To support this, we introduce the Mode Diversity loss, which promotes a spread of predictions and is depicted in Figure 2c, that favours more spread predictions over the concentrated ones, as long as the predictions are feasible. This function first excludes any trajectories that are off-road by employing an indicator function $\mathbb{1}(i)$ which assesses whether trajectory $\boldsymbol{y}^i$ remains within drivable area. The diversity loss is then calculated as the sum of pairwise distances between all remaining feasible trajectories:

$$\text{Diversity}(y, \mathbb{1}) = \frac{1}{\binom{M}{2}} \sum_{\substack{1 \leq i \leq M \\ \mathbb{1}(i)=1}} \sum_{\substack{i \leq j \leq M \\ \mathbb{1}(j)=1}} \frac{1}{T} \sum_{t=1}^{T} \|\boldsymbol{y}_t^i - \boldsymbol{y}_t^j\|_2, \tag{4}$$

where $\mathbb{1}(i) = 1$ indicates that trajectory $\boldsymbol{y}^i$ is within drivable area. This selective approach ensures that the Mode Diversity loss only considers trajectories that are practical and safe, thereby avoiding an undesired increase in the loss value from trajectories pushed outside of road boundaries.

### 3.4 ADAPTIVE LOSS WEIGHTING

Using the three proposed auxiliary loss functions requires assigning appropriate weights to balance their contributions during training. Manually tuning these weights is computationally expensive and time-consuming, as it introduces additional hyperparameters that must be optimized. Moreover, keeping a fixed weight throughout training is suboptimal, as the importance of each auxiliary loss can vary over time. To address these challenges, we introduce an adaptive loss weighting strategy that dynamically adjusts the contribution of each auxiliary loss, ensuring they provide meaningful supervision without degrading model accuracy.

The overall training objective is formulated as:

$$\mathscr{L}_{\text{final}} = \mathscr{L}_{\text{original}} + \sum_{j=1}^{3} w_j(n) \mathscr{L}_{\text{aux}_j}, \tag{5}$$

where $n$ denotes the training iteration step, and $w_j(n)$ represents the weight assigned to each auxiliary loss function at iteration $n$. Our goal is to determine functions $w_j(n)$ that adaptively scale the auxiliary losses to optimize training stability and performance.

Drawing inspiration from prior work in multi-loss learning Yu et al. (2020); Chen et al. (2018); Du et al. (2018) and based on our own observations, we identify two key challenges in combining loss functions:

1. **Gradient Conflicts:** If the gradient of an auxiliary loss is misaligned with the gradient of the main loss, it may oppose the optimization direction, reducing model accuracy. This can be quantified by the cosine similarity:

$$S_j(n) = \frac{\nabla \mathscr{L}_{\text{original}} \cdot \nabla \mathscr{L}_{\text{aux}_j}}{\|\nabla \mathscr{L}_{\text{original}}\| \|\nabla \mathscr{L}_{\text{aux}_j}\|}. \tag{6}$$

A negative $S_j(n)$ indicates that optimizing $\mathscr{L}_{\text{aux}_j}$ directly conflicts with the primary objective.

2. **Gradient Magnitude Imbalance:** If the norm of an auxiliary loss gradient is significantly larger than that of the main loss, it can dominate optimization, even when they are aligned in direction.

To mitigate these issues, we propose an adaptive weighting mechanism that dynamically adjusts auxiliary loss contributions at each training step. Specifically, we compute an estimated weight for each auxiliary loss:

$$\hat{w}_j(n) = \frac{\|\nabla \mathscr{L}_{\text{original}}\|}{\|\nabla \mathscr{L}_{\text{aux}_j}\|} \cdot S_j(n), \tag{7}$$

In this formulation, the first term scales the loss so that its gradient matches the magnitude of the original loss gradient, preventing it from dominating training. Meanwhile, the cosine similarity $S_j(n)$ suppresses auxiliary losses that conflict with the original loss by assigning negative or small values when needed.

As gradient norms and cosine similarities can be noisy, we smooth the weight updates using an exponentially weighted moving average:

$$w_j(n) = \eta w_j(n-1) + (1-\eta)\hat{w}_j(n), \tag{8}$$

where $\eta$ is a smoothing factor that controls how quickly the weights adapt to changes in gradient behavior.

If an auxiliary loss consistently conflicts with the original loss over multiple iterations, its associated weight may become negative. To prevent this from negatively impacting the model, such as encouraging off-road predictions, we set the weight to zero when applying it to the loss function while still allowing negative values during weight updates for stability. In addition, we find that in the early stages of the training, the gradients are not very reliable, which could lead to unrealistic loss weights and eventually hurt model convergence. We therefore find it crucial to use a warm-up approach for the first few epochs of training where we only update $w_j(n)$ without applying the auxiliary loss functions.

## 4 EXPERIMENTS

In this section, we detail our experimental setup, including a description of the vehicle trajectory prediction datasets and the baseline models utilized. We enhanced the baseline models by integrating our proposed loss functions as auxiliary components, and weight them using our proposed adaptive weighting strategy. Notably, we find that our algorithm is quite robust in determining effective loss weights, and the same $\eta = 0.01$ is used in all our experiments expanding two baselines and two datasets.

We present both quantitative results demonstrating the performance improvements achieved with our auxiliary loss functions and qualitative examples that illustrate specific enhancements in trajectory prediction thanks to our methodology. We show the importance and positive effect of using all our proposed loss functions together. Additionally, we show the improved robustness of our models through evaluations using the SceneAttack benchmark Bahari et al. (2022) in scenarios that include synthetically introduced turns. Finally, we share the evolution of the weights of our loss functions during training, which are adjusted to the model using our adaptive weighting scheme. We also analyze the sensitivity of our loss function weights in §C.

### 4.1 EXPERIMENTAL SETUP

Our experiments leverage the UniTraj framework Feng et al. (2024) to integrate our proposed loss functions with two state-of-the-art trajectory prediction models: Wayformer Nayakanti et al. (2022) and AutoBots Girgis et al. (2022). Wayformer is known for its superior prediction capabilities within UniTraj, whereas AutoBots provides a performant yet lightweight alternative. The experiments are conducted on two prominent datasets: nuScenes Caesar et al. (2019), a smaller and more challenging dataset, and Argoverse 2 Wilson et al. (2021), which offers a broader range of scenarios.

We adopt UniTraj's setup, where the models are trained using a history of 2 seconds and make predictions over 6 seconds, with each model outputting $M = 6$ possible trajectories, using UniTraj's model-specific hyperparameters. We report a comprehensive list of hyper parameters used in our experiments in §E for reproducibility.

In terms of evaluation metrics, we introduce three novel measures — Offroad, Direction Error, and Diversity — to assess specific aspects of prediction quality. Additionally, we utilize standard metrics

from the field which evaluate prediction accuracy, including minimum Average Displacement Error (minADE), as previously defined in Equation (1), along with minimum Final Displacement Error (minFDE) and Miss Rate (MR). MinFDE is calculated as:

$$\text{minFDE} = \min_{1 \le m \le M} \|\boldsymbol{y}_T^m - \hat{\boldsymbol{y}}_T^m\|_2. \tag{9}$$

Miss Rate (MR) is defined as the ratio of the samples where the minFDE exceeds 2 meters, and is useful where deviations up to 2 meters are acceptable.

Table 1: Quantitative results on nuScenes and Argoverse 2. Our proposed method (+ All) with adaptive loss weighting improves safety (Offroad, Direction) and diversity while maintaining accuracy (minADE, minFDE, MR). Ablations show that each loss enhances its targeted metric while also benefiting others, with the best results achieved when all are combined. Comparisons with other weighting strategies demonstrate the success of our approach in balancing auxiliary losses without harming performance, which others fail to do.

| | nuScenes | | | | | | Argoverse 2 | | | | | |
| | minADE↓ | minFDE↓ | MR↓ | Offroad↓ | Direction↓ | Diversity↑ | minADE↓ | minFDE↓ | MR↓ | Offroad↓ | Direction↓ | Diversity↑ |
|---|---|---|---|---|---|---|---|---|---|---|---|---|
| Ground Truth | 0 | 0 | 0 | 0.12 | 4.15 | - | 0 | 0 | 0 | 0.16 | 2.73 | - |
| Wayformer | 1.09 | 2.53 | 0.45 | 2.72 | 7.66 | 3.82 | 0.86 | 1.76 | 0.28 | 0.59 | 4.18 | 3.82 |
| + Offroad | 1.09 | 2.53 | 0.42 | 1.50 | 5.96 | 3.83 | 0.86 | 1.76 | 0.28 | 0.37 | 3.83 | 3.85 |
| + Direction | 1.09 | 2.53 | 0.43 | 2.11 | 6.08 | 3.83 | 0.86 | 1.76 | 0.29 | 0.48 | 3.04 | 3.82 |
| + Diversity | 1.10 | 2.56 | 0.44 | 2.92 | 7.88 | 3.85 | 0.86 | 1.76 | 0.28 | 0.59 | 4.18 | 3.84 |
| + All (ours) | 1.10 | 2.56 | 0.42 | **1.36** | **5.14** | **4.73** | 0.86 | 1.75 | 0.28 | **0.35** | 3.01 | 3.87 |
| + All (HP) | 1.13 | 2.61 | 0.43 | 1.67 | 5.47 | 4.47 | 0.89 | 1.85 | 0.30 | 0.44 | **2.54** | **4.2** |
| + All (Cosine) | 1.24 | 2.95 | 0.50 | 1.54 | 5.98 | 15.24 | 0.98 | 2.19 | 0.41 | 1.07 | 5.74 | 36.67 |
| + All (GradNorm) | 2.74 | 7.51 | 0.86 | 26.51 | 706.24 | 52.05 | 1.83 | 2.99 | 0.61 | 2.52 | 7.36 | 7.06 |
| Autobots | 1.27 | 2.64 | 0.42 | 1.83 | 5.99 | 4.46 | 0.85 | 1.68 | 0.26 | 0.32 | 3.55 | 3.83 |
| + Offroad | 1.27 | 2.67 | 0.42 | 1.16 | 5.10 | 4.53 | 0.85 | 1.68 | 0.26 | 0.23 | 3.57 | 3.89 |
| + Direction | 1.26 | 2.63 | 0.42 | 1.23 | 4.84 | 4.56 | 0.85 | 1.69 | 0.26 | 0.27 | 2.47 | 3.88 |
| + Diversity | 1.27 | 2.64 | 0.43 | 1.82 | 6.03 | 4.51 | 0.85 | 1.69 | 0.26 | 0.30 | 3.55 | 3.83 |
| + All (ours) | 1.26 | 2.63 | 0.42 | **0.88** | 4.48 | 4.78 | 0.85 | 1.70 | 0.26 | **0.22** | 2.47 | **3.89** |
| + All (HP) | 1.26 | 2.70 | 0.43 | 1.01 | **4.27** | **4.93** | 0.91 | 1.90 | 0.31 | 0.20 | 1.90 | 4.27 |
| + All (Cosine) | 1.27 | 2.67 | 0.43 | 1.73 | 5.93 | 4.38 | 0.86 | 1.68 | 0.26 | 0.30 | 3.41 | 3.86 |
| + All (GradNorm) | 3.07 | 7.54 | 0.83 | 23.76 | 1068.54 | 28.88 | 2.28 | 5.55 | 0.74 | 5.06 | 106.40 | 83.51 |

## 4.2 QUANTITATIVE RESULTS

The quantitative performance of our method, detailed in Tab. 1, demonstrates significant improvements in safety and scene-understanding metrics while maintaining accuracy. Our proposed approach, which integrates all three auxiliary loss functions with the adaptive loss weighting strategy, achieves the most consistent enhancements across both datasets and two baseline models.

**Ablation of Auxiliary Losses:** To analyze the contribution of each auxiliary loss function, we conduct an ablation study, training models with each loss function in isolation. The results show that each auxiliary loss improves performance in its targeted metric while also positively influencing others. For instance, Offroad and Direction Consistency Losses benefit each other, as being closer and more aligned to centerline points also reduces offroad metric. However, the most pronounced improvements occur when all losses are applied together, particularly in diversity, which sees only limited gains when used alone but surprisingly improves substantially when combined with the other losses, which highlights complementary effects of the proposed loss functions.

**Comparison of Adaptive Loss Weighting Strategies:** We also compare our adaptive loss weighting strategy against alternative baselines:
**+ All (HP):** This variant relies on manual hyperparameter tuning to adjust loss weights for each model and dataset. While it sometimes achieves stronger improvements in scene-compliance metrics, it struggles to maintain accuracy and requires extensive tuning, making it computationally expensive and impractical.
**+ All (Cosine Du et al. (2018)):** This approach disables auxiliary loss functions when their gradients have a negative cosine similarity to the original loss. However, it fails to account for gradient magnitude, often leading to accuracy degradation and suboptimal performance.
**+ All (GradNorm Chen et al. (2018)):** This method adjusts loss weights based solely on gradient norms, without considering conflicts between losses. As a result, it assigns unrealistic weights, destabilizing training and leading to poorly optimized models.

Overall, our adaptive loss weighting strategy balances auxiliary losses effectively, achieving stronger safety and robustness improvements while preserving accuracy, making it a practical and scalable solution.

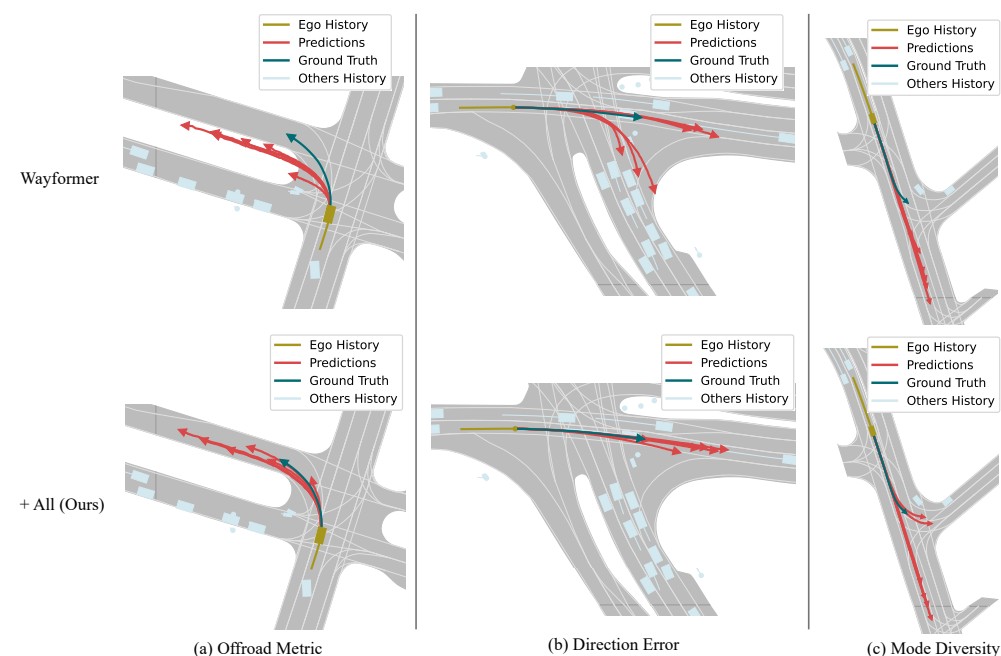

(a) Offroad Metric      (b) Direction Error      (c) Mode Diversity

Figure 3: Comparison of Wayformer predictions when trained with its original loss function on the first row, and when it is augmented with our auxiliary loss package in row two. Each column highlights a specific improvement: (a) Offroad Loss corrects off-road predictions by enhancing adherence to drivable areas; (b) Direction Error adjusts potentially hazardous right turns against traffic flow; (c) Mode Diversity introduces a new left turn which models the ground truth trajectory better. More qualitative results could be found in §F.

## 4.3 QUALITATIVE RESULTS

We present qualitative results of our proposed auxiliary loss functions in Figure 3, demonstrating significant improvements in model predictions across various scenarios. In the first panel, the baseline model generates multiple off-road predictions, failing to align with the ground truth. Incorporating the Offroad loss during training teaches the model to recognize and avoid such errant paths, leading to more accurate trajectory predictions. In the second panel, a notable error in the baseline predictions includes a dangerous right turn against traffic flow. The application of our Direction Error loss corrects this, aligning the predictions with the correct traffic direction, thus enhancing safety and compliance with traffic rules. The third panel showcases the effects of our Mode Diversity loss, which significantly increases the spread of predicted trajectories. This loss function not only introduces additional plausible maneuvers, such as a new left turn that was absent in the baseline predictions but also ensures that the predicted trajectories are better spaced, reducing the likelihood of missing behaviors and increasing the overall prediction diversity.

Table 2: Offroad metrics in meters, under Scene Attack Bahari et al. (2022). Adding naturalistic turns increases offroad errors, especially for baseline models. Despite not being trained on such distribution shifts, models trained with our auxiliary losses exhibit stronger resilience, with the highest improvements observed when using all losses together.

| | nuScenes | | Argoverse 2 | |
|---|---|---|---|---|
| | Original | Attacked | Original | Attacked |
| Wayformer | 2.72 | 8.71 | 0.59 | 3.48 |
| + Offroad | 1.50 | 6.79 | 0.37 | 2.86 |
| + All | **1.36** | **6.57** | **0.35** | **2.75** |
| Autobots | 1.83 | 4.30 | 0.32 | 1.28 |
| + Offroad | 1.16 | 3.34 | 0.23 | 1.08 |
| + All | **0.88** | **2.98** | **0.22** | **0.98** |

## 4.4 ROBUSTNESS TO SCENE ATTACK

We evaluate the robustness of our models using the Scene Attack benchmark Bahari et al. (2022), which introduces naturalistic perturbations by modifying road geometry ahead of the ego vehicle, such as adding or altering turns. Importantly, our models were not trained on such distribution shifts, making this an effective test of their generalization ability.

Table 2 reports the Offroad metric for baseline models and our improved versions across the Argoverse 2 and nuScenes datasets. As expected, all models have higher offroad rates when exposed to these scene perturbations. However, models trained with our auxiliary loss functions consistently show better resilience, achieving significantly lower offroad errors even in unseen, manipulated environments. These improvements are most pronounced when using all loss functions together, demonstrating their complementary effect in enforcing road adherence and scene awareness.

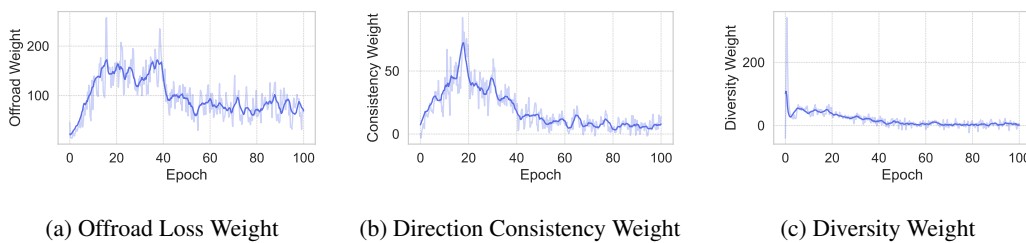

(a) Offroad Loss Weight     (b) Direction Consistency Weight     (c) Diversity Weight

Figure 4: Evolution of adaptive loss weights during training on nuScenes using AutoBots with all proposed loss functions. Each plot illustrates how the assigned weight for (a) Offroad Loss, (b) Direction Consistency Loss, and (c) Diversity Loss evolves over 100 epochs. The weights adjust dynamically based on gradient norms and similarity, reflecting the changing importance of each auxiliary loss function throughout training.

## 4.5 ADAPTIVE LOSS WEIGHT EVOLUTION

We visualize the evolution of loss weights in Figure 4, showing how our adaptive weighting method adjusts the importance of each auxiliary loss during training of AutoBots on nuScenes. Initially, Offroad Loss and Direction Consistency Loss receive low weights, as early predictions are small and rarely go off-road. Their weights increase as predictions expand, enforcing road adherence, before stabilizing at lower values in later training stages. Conversely, Diversity Loss starts with a high weight to encourage trajectory spread when predictions are initially condensed around the last position of the ego vehicle, then gradually decreases as diversity naturally improves.

## 5 CONCLUSIONS

We presented a unified package of auxiliary objectives—Offroad Loss, Direction Consistency Loss, and Diversity Loss—that strengthen trajectory predictors by enforcing road compliance, traffic flow consistency, and multimodal coverage. While each objective is useful on its own, our results show that their combination, together with adaptive weighting, is key: it consistently improves safety and robustness without sacrificing accuracy. Evaluations on nuScenes and Argoverse 2 confirm large reductions in off-road violations, and under the challenging SceneAttack benchmark we show that learned road compliance transfers to unseen out-of-distribution scenarios. Our approach is lightweight, requires no manual tuning, and can be plugged into existing models such as Wayformer and AutoBots.

These findings highlight the value of combining inductive biases with adaptive weighting for reliable trajectory forecasting. Promising directions for future work include extending the auxiliary suite with objectives such as collision avoidance or kinematic feasibility, and refining adaptive weighting to adapt to driving context—for example, prioritizing road adherence in structured urban layouts while encouraging greater diversity in uncertain settings.

## 6 REPRODUCIBILITY STATEMENT

We will release our code to facilitate reproducibility of all experiments. Key hyperparameters are reported in §4 and §4.1, with a complete list provided in §E.

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

## A  FREQUENTLY ASKED QUESTIONS

**Does your method improve accuracy of prediction models?**
Not significantly. Our method is focused on improving other aspects of model performance, such as scene understanding and diversity. Although we sometimes improve accuracy marginally, our primary aim is to improve model robustness and generalization. Please keep in mind that the baseline methods are trained to optimize accuracy, therefore, by adding other objectives to minimize, it is expected to have degradation in terms of accuracy, which our adaptive weight strategy successfully controls.

**Are your loss functions differentiable?**
Yes! We carefully designed all our loss functions to be differentiable, and their gradients guiding model towards being safer and more diverse. This allows us to use normal gradient-based optimizations, unlike other works Casas et al. (2020) which use reinforcement learning to optimize their non-differentiable loss functions. This helps our model to be more easily adaptable to various prediction models, while also enables our gradient-based adaptive weighting strategy.

**Does your method increase computational complexity of the model?**
Since our method proposes an objective function package, it does not alter the model architecture in any way, hence the computation complexity of our models, specifically during inference, remains the same. During training however, we need multiple backward passes through the model for computing the gradient of our different loss functions. Therefore our training times are longer than normal training when using our adaptive weighting method. This holds for all the previous works Du et al. (2018); Chen et al. (2018); Yu et al. (2020) which also require gradient of different auxiliary loss functions. There are ways to reduce this training overhead, *e.g.* by calculating the gradient only with respect to a subset of model parameters. We did not explore this avenue, as the training times of trajectory prediction models are not very long, and each model training would finish in less than 24 hours, even with multiple backward passes through the model, in the worst case. For other domains where the increase in training time could be an issue, another way to mitigate multiple backward passes at each iteration is to reduce the frequency of updating the auxiliary loss weights, *e.g.* instead of updating loss weights after each iteration, do it every 20 iterations, which should reduce the overhead by a high margin and is not expected to degrade the results.

## B  LARGE LANGUAGE MODEL USAGE STATEMENT

We used large language models to refine the writing of text and tables, and as an assistant in developing parts of the model code.

## C  LOSS WEIGHT STUDY

In this study, we explore how combining the original loss function with our auxiliary loss affects model performance. We illustrate this relationship in Figure 6, where we adjust the weight of the auxiliary loss, $\alpha$, when added to the original loss function of the model:

$$\mathscr{L}_{\text{final}} = \mathscr{L}_{\text{original}} + \alpha\mathscr{L}_{\text{auxiliary}}. \tag{10}$$

Starting with a high auxiliary weight $\alpha$, we exponentially decrease this weight, moving towards the red baseline point, which typically worsens the auxiliary metrics but can improve the main prediction accuracy.

Interestingly, there is a sweet spot along these curves where the prediction accuracy, specifically minADE, is close to that of the baseline model, while simultaneously improving on the auxiliary metrics. This balance demonstrates that it's possible to enhance certain aspects of a model's predictions without degrading overall accuracy. Moreover, these results show that our loss functions are not too sensitive to these weights.

## D  SIGNED DISTANCE FUNCTION ALGORITHM

---

**Algorithm 1** Computation of Signed Distance Function $\phi(\boldsymbol{p}, \Omega)$

---

**Require:** Point $\boldsymbol{p}$, Polygon $\Omega = \{e_1, e_2, \ldots, e_N\}$ with edges $e_i$
**Ensure:** Signed distance $\phi(\boldsymbol{p}, \Omega)$
 1: **Initialize** $d_{\min} \leftarrow \infty$                              ▷ Track min distance to edges
 2: **for** each edge $e_i = (\boldsymbol{v}_i, \boldsymbol{v}_{i+1}) \in \Omega$ **do**
 3:      $d_i \leftarrow \text{DistanceToEdge}(\boldsymbol{p}, e_i)$
 4:      **if** $d_i < d_{\min}$ **then**
 5:          $d_{\min} \leftarrow d_i$                              ▷ Update closest boundary distance
 6:      **end if**
 7: **end for**
                                                                   ▷ Ray casting for point-in-polygon test
 8: **Initialize** $crossings \leftarrow 0$
 9: **for** each edge $e_i = (\boldsymbol{v}_i, \boldsymbol{v}_{i+1}) \in \Omega$ **do**
10:      **if** RayIntersectsEdge$(\boldsymbol{p}, e_i)$ **then**
11:          $crossings \leftarrow crossings + 1$
12:      **end if**
13: **end for**
14: **if** $crossings \mod 2 = 1$ **then**
15:      $\phi(\boldsymbol{p}, \Omega) \leftarrow -d_{\min}$                              ▷ Point is inside
16: **else**
17:      $\phi(\boldsymbol{p}, \Omega) \leftarrow d_{\min}$                              ▷ Point is outside
18: **end if**
19: **return** $\phi(\boldsymbol{p}, \Omega)$

---

## E  HYPER PARAMETERS

In this section we share the hyper parameters we used in our experiments. We only report the hyper parameters designed by our methodology, and refer the readers to the respective papers and UniTraj Feng et al. (2024) for hyper parameters of prediction models.

For our Offroad Loss, we use a margin of $m = 0.5$ meters. Direction Consistency also has a distance margin of $m_d = 2$ meters and an angular margin of $m_\theta = \frac{\pi}{3}$ radians. Finally, we would filter out prediction before calculating the Diversity loss, if their Offroad metric is larger than 2 meters.

As also stated in §4, we set the smoothing factor of our adaptive weighting scheme to $\eta = 0.01$ for all our experiments. We do warmup for AutoBots for 5 epochs, and Wayformer for 10 epochs, where we do not take into effect our auxiliary loss functions. For fairness, we do the same warmup strategy for the other methods we compare to in Tab. 1.

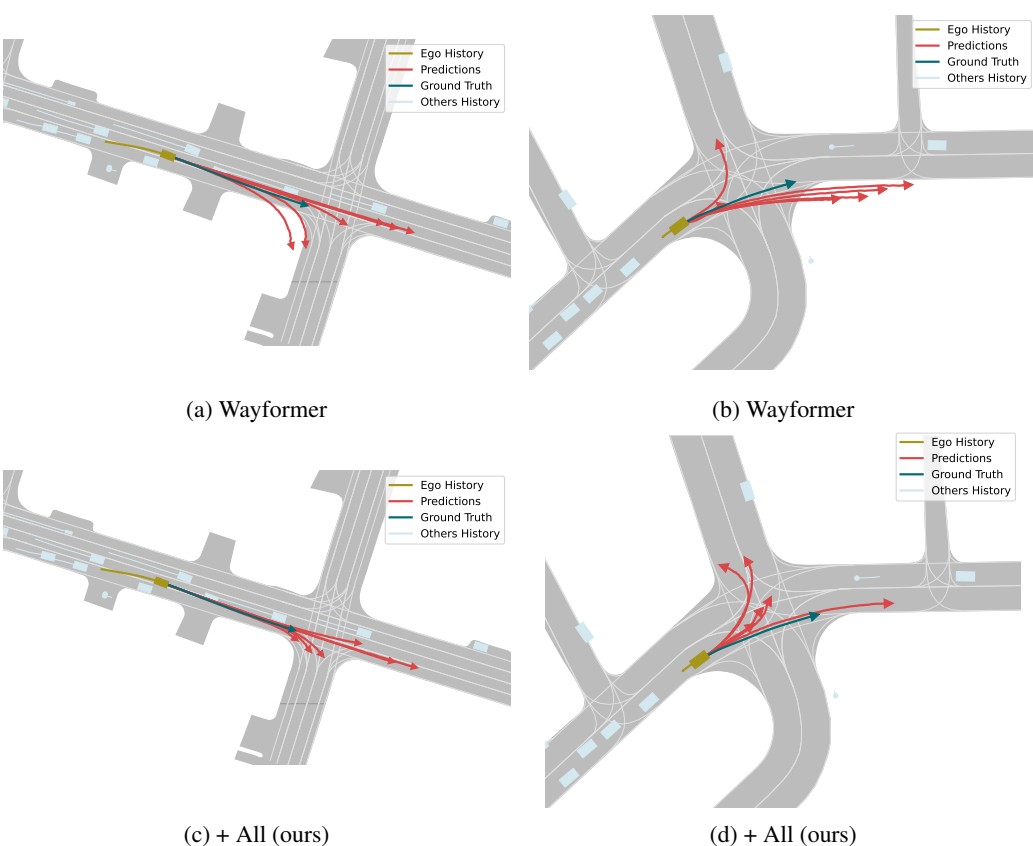

(a) Wayformer

(b) Wayformer

(c) + All (ours)

(d) + All (ours)

Figure 5: Qualitative results of Offroad Metric improvement.

# F MORE QUALITATIVE EXAMPLES

In this section, we present more qualitative examples of improvement of predicted trajectories in various aspects, thanks to adapting our proposed loss functions. Figure 5 and Figure 7 show more qualitative examples where our proposed approach reduces Offroad error of the model. Figure 8 presents more scenarios where Wayformer baseline has dangerous predictions opposite the flow of traffic, while our proposed method rectifies this issue. Finally, Figure 9 and Figure 10 present more examples where integration of all our proposed loss functions enhances mode diversity in models predictions.

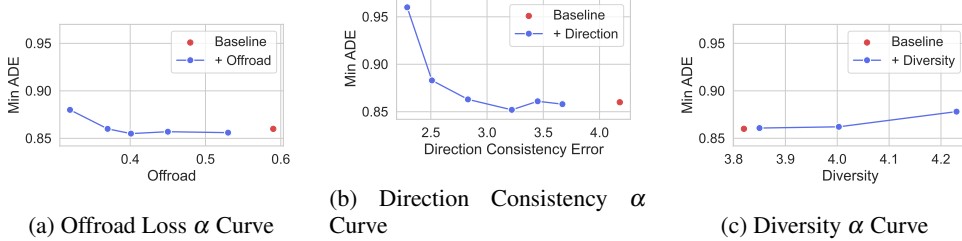

(a) Offroad Loss $\alpha$ Curve

(b) Direction Consistency $\alpha$ Curve

(c) Diversity $\alpha$ Curve

Figure 6: Performance impact of integrating our loss functions on Wayformer's minADE across the Argoverse 2 dataset. The blue curves demonstrate the trade-off between increasing auxiliary loss weights and prediction accuracy, with red dots marking the baseline performance. There exist regions where enhanced losses maintain similar accuracy compared to the baseline. Similar patterns are observed across different models and dataset configurations.

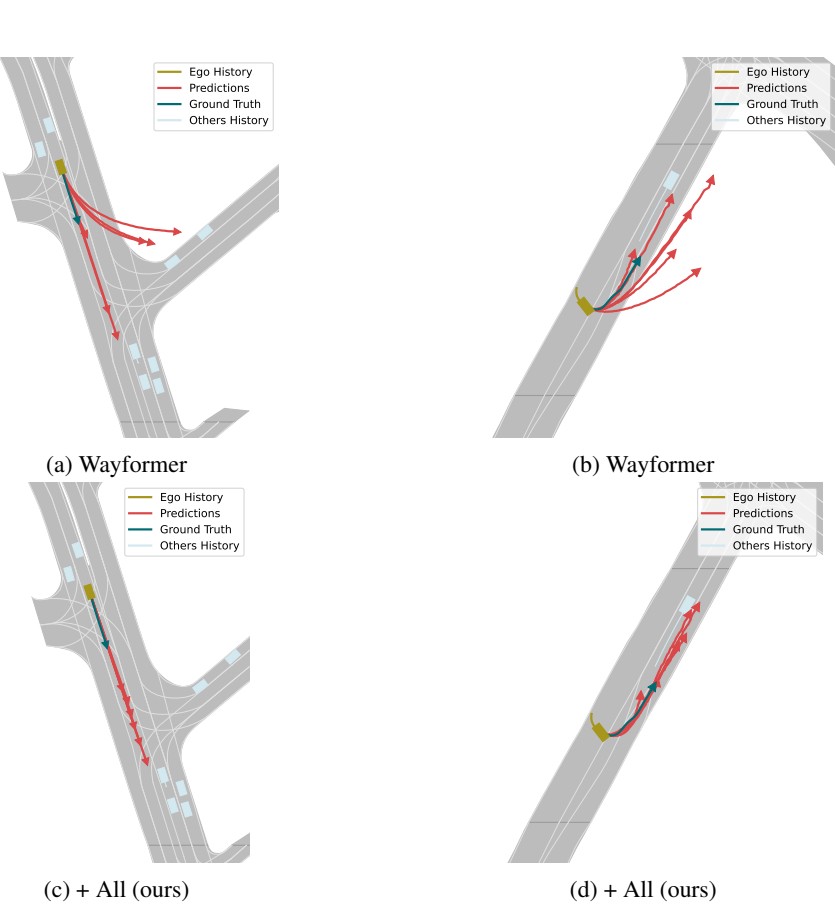

Figure 7: Qualitative results of Offroad Metric improvement.

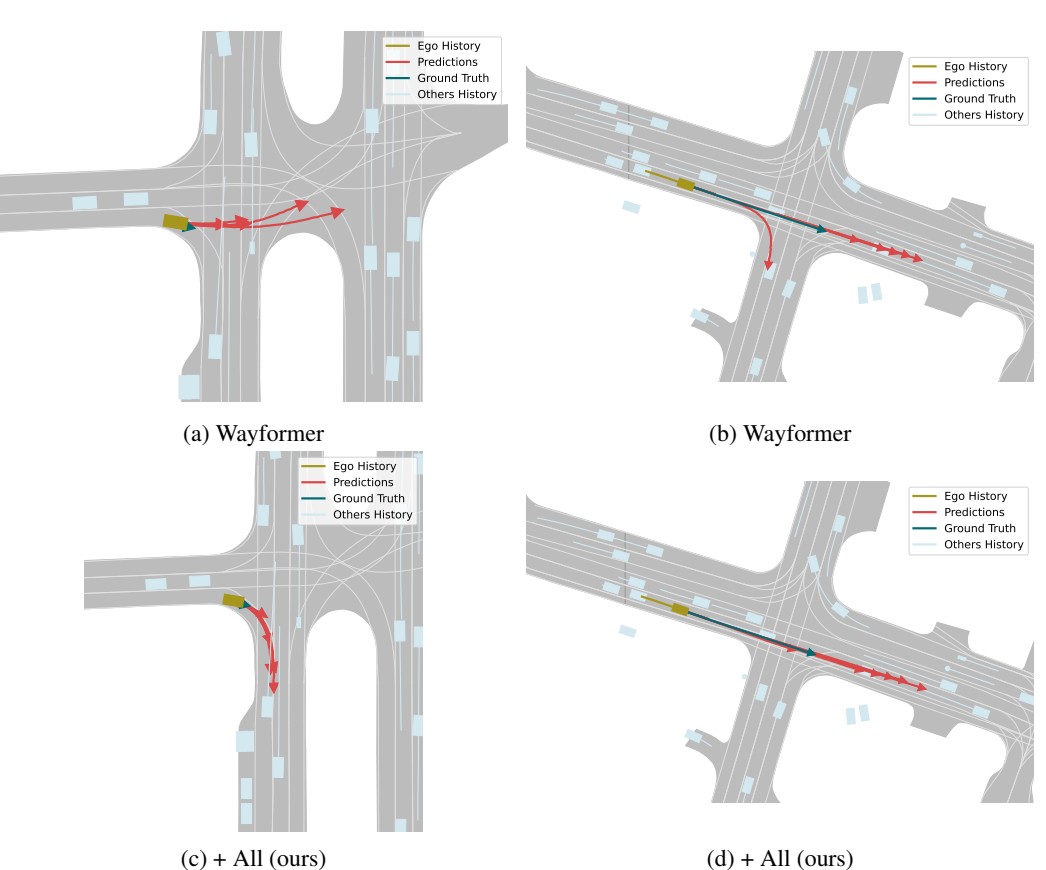

Figure 8: Qualitative results of Direction Consistency improvement.

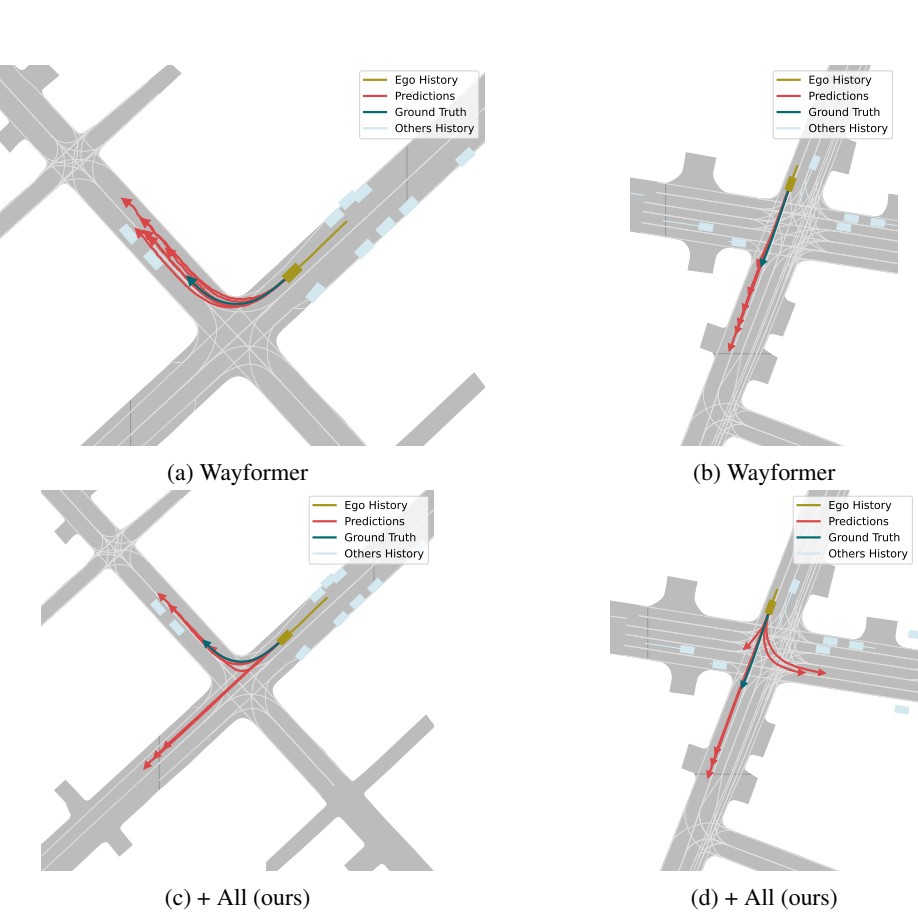

(a) Wayformer

(b) Wayformer

(c) + All (ours)

(d) + All (ours)

Figure 9: Qualitative results of Mode Diversity improvement.

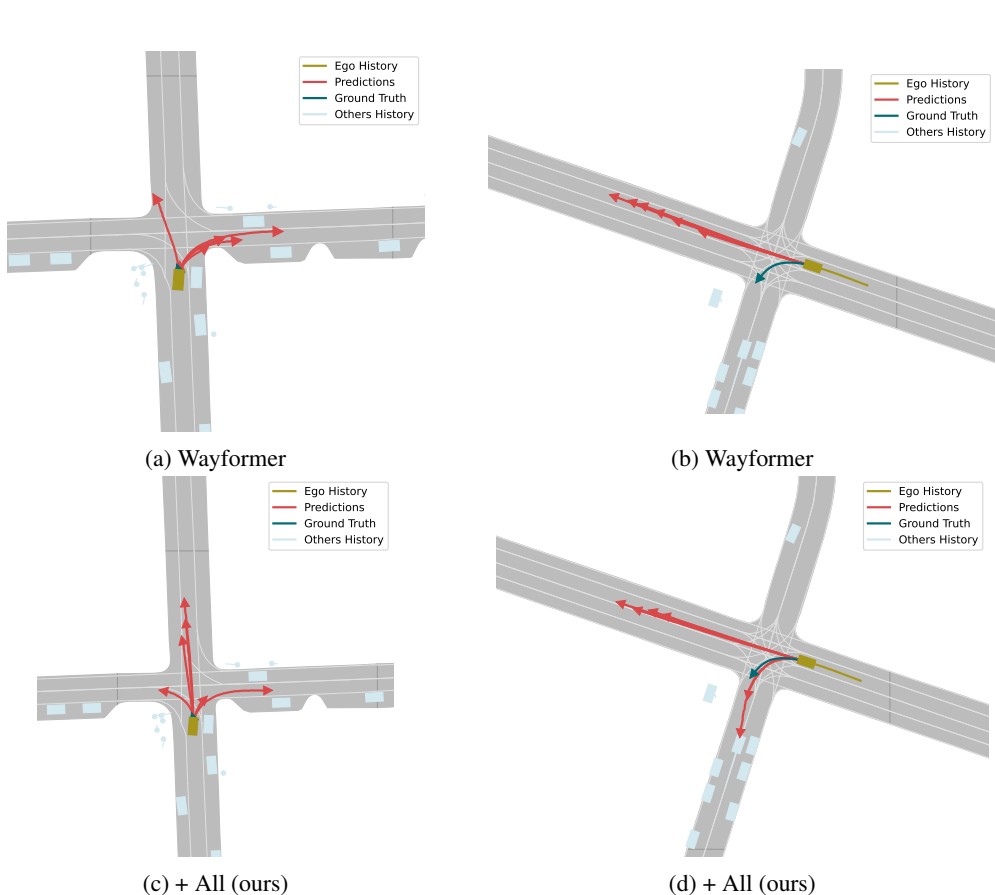

(a) Wayformer

(b) Wayformer

(c) + All (ours)

(d) + All (ours)

Figure 10: Qualitative results of Mode Diversity improvement.

