# OpenReview forum: "Combining Auxiliary Losses for Safer and More Robust Trajectory Prediction"
_ICLR.cc/2026/Conference — Submitted to ICLR 2026_

### Official Review · Reviewer_uBa8 · 2025-10-30

**Soundness:** 2
**Presentation:** 2
**Contribution:** 2
**Rating:** 4
**Confidence:** 4

**Summary:**

This paper proposes a unified auxiliary loss package to improve the safety and robustness of vehicle trajectory prediction models. The authors identify that standard accuracy-focused losses (like minADE) often lead to predictions that are off-road, violate traffic direction, or lack diversity. To address this, the paper introduces three auxiliary losses: Offroad Loss, Direction Consistency Loss, and Diversity Loss.

A key finding is that while each loss improves its targeted metric, their combination is essential for robust performance. To balance these losses with the main objective without costly manual tuning, the authors propose an adaptive weighting scheme. This technique dynamically adjusts the weight of each auxiliary loss based on its gradient's magnitude and its cosine similarity with the main loss gradient.

The method is evaluated by integrating it into two baseline models (Wayformer and AutoBots) on the nuScenes and Argoverse 2 datasets. Results show consistent improvements in safety metrics and diversity without sacrificing accuracy. The paper also demonstrates improved robustness on the SceneAttack benchmark, showing a 25% reduction in off-road errors in out-of-distribution scenarios.

**Strengths:**

1. The paper tackles a clear and important problem in trajectory prediction: the lack of scene compliance in state-of-the-art models. The focus on safety-oriented metrics (off-road, direction) is highly relevant for real-world autonomous driving applications.

2. The three heuristic losses are intuitive and address distinct, critical shortcomings. Empirical findings show that these loss functions contribute to the quality of trajectory prediction. The proposed loss-weighting technique is also novel and effective in the trajectory prediction literature.

**Weaknesses:**

1. The high-level idea of using auxiliary losses to enforce scene compliance is not new. Planning benchmarks like NAVSIM[1] evaluate trajectories with multiple open-loop objectives such as drivable area compliance and no at-fault collisions, which are very similar to the losses proposed in the paper. Previous literature like Hydra-MDP also trains a neural network with such loss functions.

2. The experimental comparisons are primarily against the base versions of Wayformer and AutoBots before adding the proposed losses. There is a lack of comparison against other recent methods. For instance, the method is not compared to approaches like QCNet[3] or DeMo[4], making it difficult to evaluate its superiority over the entire field.

3. There is a minor issue in formatting. When the citation is not a part of the sentence, it should be included in the brackets (\citep instead of \citet).

[1] Dauner D, Hallgarten M, Li T, et al. Navsim: Data-driven non-reactive autonomous vehicle simulation and benchmarking[J]. Advances in Neural Information Processing Systems, 2024, 37: 28706-28719.

[2] Li Z, Li K, Wang S, et al. Hydra-mdp: End-to-end multimodal planning with multi-target hydra-distillation[J]. arXiv preprint arXiv:2406.06978, 2024.

[3] Zhou Z, Wang J, Li Y H, et al. Query-centric trajectory prediction[C]//Proceedings of the IEEE/CVF conference on computer vision and pattern recognition. 2023: 17863-17873.

[4] Zhang B, Song N, Zhang L. Decoupling motion forecasting into directional intentions and dynamic states[J]. Advances in Neural
Information Processing Systems, 2024, 37: 106582-106606.

**Questions:**

Please refer to the weaknesses section.

---

### Official Review · Reviewer_ZZZB · 2025-10-31

**Soundness:** 2
**Presentation:** 2
**Contribution:** 3
**Rating:** 4
**Confidence:** 4

**Summary:**

This paper presents a method for improving trajectory prediction by combining auxiliary losses (Offroad, Direction Consistency, and Diversity) with an adaptive weighting scheme. While the idea of systematically integrating such losses is practical and shows promise, I have several major concerns regarding the methodological assumptions, experimental rigor, and overall contribution that prevent me from supporting acceptance in its current form.

**Strengths:**

The combined auxiliary loss with adaptive weighting proposed in the paper has conceptual and engineering merit and shows improvement in several metrics.

**Weaknesses:**

While the idea of combining auxiliary losses for trajectory prediction is practical and addresses relevant challenges in scene compliance, I have several concerns regarding the technical contribution, experimental rigor, and real-world applicability of the approach.

The method systematizes existing auxiliary concepts, such as off-road penalties, direction alignment, and diversity promotion, rather than introducing fundamentally new loss formulations. The adaptive weighting scheme, though intuitive, lacks strong theoretical motivation and seems potentially sensitive to gradient noise. The authors do not sufficiently justify why their proposed weighting strategy, which combines gradient norm scaling with cosine similarity, outperforms more established multi-task learning techniques. A deeper comparative analysis with recent adaptive loss balancing methods would strengthen this claim.

A key limitation is the method's reliance on high-quality map inputs. Both the Off-road and Direction losses assume accurate drivable-area polygons and centerlines, yet real-world maps often contain noise, missing lanes, or outdated geometry. The paper does not evaluate performance under such realistic conditions, which limits the claimed practical applicability. Additionally, while the Diversity loss filters out off-road predictions to maintain feasibility, this filtering itself depends on the Off-road loss, creating a potential circular dependency that may not hold with imperfect maps.

The experimental evaluation has several shortcomings. The proposed Offroad, Direction, and Diversity metrics, though useful for quantifying specific behaviors, are self-defined and lack established baselines or clear connections to real-world safety outcomes such as collisions or planning failures. Their implementation details and threshold selections are not thoroughly explained, affecting reproducibility.

The comparison with adaptive weighting baselines raises concerns, particularly the notably poor performance of GradNorm, which may indicate implementation or hyperparameter issues. The authors should clarify whether these baselines were reproduced faithfully or adapted from official implementations. Moreover, the paper lacks comparisons with recent SOTA trajectory predictors that also address scene compliance, limiting the claim of general applicability.

Several critical experiments are missing, including evaluations under map noise or centerline errors, repeated runs with different random seeds to assess variance, sensitivity analysis of training strategies such as warm-up and exponential moving average parameters, and closed-loop testing in realistic simulation environments. These are essential for assessing the method's robustness and readiness for deployment.

Although the authors commit to releasing code, key implementation details, such as efficient GPU computation of the signed distance function, centerline matching logic, and hyperparameter ranges for weight updates, are omitted, hindering reproducibility.

In summary, while the work offers a sensible engineering contribution and shows improvements in auxiliary metrics, it falls short in demonstrating novelty, robustness, and practical generalizability. The paper would be significantly strengthened by evaluating performance under map degradation and domain shift, providing statistical significance tests and repeated run results, including comparisons with stronger baselines, and clarifying the relationship between the proposed metrics and actual driving safety. Without these additions, I find the paper not yet ready for acceptance.

**Questions:**

See the last section.

---

### Official Review · Reviewer_1ghj · 2025-11-03

**Soundness:** 2
**Presentation:** 3
**Contribution:** 2
**Rating:** 4
**Confidence:** 4

**Summary:**

This article introduces auxiliary loss functions designed to guide trajectory prediction models to follow road constraints while maintaining diversity in predicted trajectories.  The proposed method leverages the UniTraj framework and is applied to the nuScenes and Argoverse2 datasets using AutoBot and Wayformer models. It improves off-road adherence, direction consistency, and trajectory diversity while maintaining the overall accuracy of the models. In addition, the proposed approach demonstrates stronger robustness under SceneAttack scenarios. To prevent potential degradation of performance that may arise when simply adding multiple auxiliary losses, the authors introduce an adaptive loss weighting strategy that dynamically adjusts the contribution of each auxiliary loss during training.

**Strengths:**

The proposed approach is designed to ensure that vehicle trajectories adhere to road constraints and traffic rules, thereby reducing off-road or reverse-direction predictions compared to baseline models. In multimodal trajectory prediction scenarios, the method enhances the model’s ability to generate diverse yet feasible trajectories, effectively capturing multiple drivable path options. Moreover, the adaptive loss weighting mechanism ensures that the auxiliary losses are effectively integrated during training, allowing each to contribute meaningfully without compromising overall model performance.

**Weaknesses:**

1. The evaluation methodology for offroad, direction consistency, and diversity proposed by the authors appears insufficiently described. To convincingly demonstrate the effectiveness of the proposed auxiliary losses, it would have been preferable to introduce well-defined quantitative metrics or to adopt established ones from prior work.
- Relevant existing studies that provide suitable evaluation metrics include [1][2]

2. In particular, the authors could have leveraged the model MTR available in the UniTraj framework, which they already use for experiments. Given that the original MTR paper explicitly discusses the limitation of diversity evaluation, it would have been an ideal benchmark to validate the proposed method.


[1] S. H. Park et al., Diverse and Admissible Trajectory Forecasting through Multimodal Context Understanding, ECCV 2020
[2] C. Chen et al., CRITERIA: a New Benchmarking Paradigm for Evaluating Trajectory Prediction Models for Autonomous Driving, ICRA 2024

**Questions:**

What metrics were used to evaluate Offroad, Direction, and Diversity in the qualitative results?

---

### Official Review · Reviewer_5LLf · 2025-11-04

**Soundness:** 3
**Presentation:** 2
**Contribution:** 2
**Rating:** 4
**Confidence:** 4

**Summary:**

The authors address the problem of trajectory predictors outputting unrealistic/unfeasible trajectories, focusing on reducing off-road and wrong-way predictions. In addition, they also propose to use diversity loss, to make the outputs more diverse. They explore the use of these three losses and how to combine them under a single setup, where they propose an approach to dynamically weight the losses based on their gradients and how they relate to the gradients of the main loss. They apply the new aux losses to two well-known baselines using two open-sourced data sets, and show the benefits of adding them.

**Strengths:**

- Interesting and relevant topics being investigated.
- Straightforward proposal, easy to integrate into existing methods.
- Promising results shown.

**Weaknesses:**

- Novelty is limited.
- The authors did not fully position their approaches relative to the existing work.
- Writing can be improved, some parts of the methodology are not well explained.

**Questions:**

- Many references are missing throughout the paper, where the authors provide some claims without references. E.g., line 31, "Recent advances ...".
- Line 147, "only the closest trajectory to the ground-truth receives gradient ...", reference missing. Also, there are methods that do not have this limitation (e.g., relying on a mixture of Gaussians), but they are usually not as performant. Still good to briefly discuss to provide an additional context.
- The authors claim that they propose the new losses, but some of them were already existing. E.g., off-road loss was shared before, the authors even cite a few in Section 2, but then still say that their approach is fully novel which does not seem exactly right. They should provide more context and discussion if they are to claim that.
- Line 203, "from corresponding centerline", how do you compute the corresponding one? Explanation is handwavy.
- Overall, Section 3.3 requires much more references to frame their losses into the right context. Currently there is none, and the authors claim full novelty which seems like a stretch.
- Equation 7 means that the aux loss will be multiplies even if it has very small norm? What is the potential downside of that approach? Should be discussed.
- The three novel metrics are not introduced properly, there are no explanations given.
- Similarly to the losses, these metrics also don't seem fully novel, and that should also be discussed more.
- nit: "to the original loss", should actually be "to the gradient of the original loss"?
- What are the downsides of the approach? Should be at least briefly discussed. E.g., how do the losses affect non-compliant actors, driving off-road or in the wrong direction?

---

### Meta-Review · Area_Chair_mxTN · 2026-01-07

**Summary:**

Reviewers raise concerns regarding novelty and experiments, which I concur.

No rebuttal is provided.

Thus, I recommond rejection.

**Reviewer Concerns:**

No rebuttal is provided. Thus, no changes.

**Reviewer Scores:**

No rebuttal is provided. Thus, no changes.

---

### Decision · Program_Chairs · 2026-01-26

Reject